# IAF, QGF, and QDF Peptides Exhibit Cholesterol-Lowering Activity through a Statin-like HMG-CoA Reductase Regulation Mechanism: In Silico and In Vitro Approach

**DOI:** 10.3390/ijms222011067

**Published:** 2021-10-14

**Authors:** Mariana Silva, Biane Philadelpho, Johnnie Santos, Victória Souza, Caio Souza, Victória Santiago, Jaff Silva, Carolina Souza, Francine Azeredo, Marcelo Castilho, Eduardo Cilli, Ederlan Ferreira

**Affiliations:** 1School of Pharmacy, Federal University of Bahia, Salvador 40170-115, BA, Brazil; marianabarros.cs@gmail.com (M.S.); biane_philadelpho@hotmail.com (B.P.); johnnie.machado25@gmail.com (J.S.); victoriacruz.29@outlook.com (V.S.); caioacs1@gmail.com (C.S.); victoria.santiago@hotmail.com (V.S.); jrdsilva@ufba.br (J.S.); carolods@ufba.br (C.S.); francine.azeredo@ufba.br (F.A.); castilho@ufba.br (M.C.); 2Chemistry Institute, Sao Paulo State University, Araraquara 14800-900, SP, Brazil

**Keywords:** cowpea peptides, molecular docking, pharmacokinetic properties, competitive HMG-CoA reductase inhibitor

## Abstract

In this study, in silico approaches are employed to investigate the binding mechanism of peptides derived from cowpea β-vignin and HMG-CoA reductase. With the obtained information, we designed synthetic peptides to evaluate their in vitro enzyme inhibitory activity. In vitro, the total protein extract and <3 kDa fraction, at 5000 µg, support this hypothesis (95% and 90% inhibition of HMG-CoA reductase, respectively). Ile-Ala-Phe, Gln-Gly-Phe, and Gln-Asp-Phe peptides were predicted to bind to the substrate binding site of HMGCR via HMG-CoAR. In silico, it was established that the mechanism of HMG-CoA reductase inhibition largely entailed mimicking the interactions of the decalin ring of simvastatin and via H-bonding; in vitro studies corroborated the predictions, whereby the HMG-CoA reductase activity was decreased by 69%, 77%, and 78%, respectively. Our results suggest that Ile-Ala-Phe, Gln-Gly-Phe, and Gln-Asp-Phe peptides derived from cowpea β-vignin have the potential to lower cholesterol synthesis through a statin-like regulation mechanism.

## 1. Introduction

Global data suggest that non-communicable diseases (NCDs) are the leading cause of death worldwide and among the most challenging public health issues of the 21st century [1]. On the one hand, the increased prevalence of NCDs results from improved treatments of infections and other lethal diseases, economic growth, and urbanisation, which, consequently, increases life expectancy worldwide. On the other hand, psychological stress and hereditary factors also contribute to high levels of NCDs [2].

According to the World Health Organization (WHO), cardiovascular diseases (CVDs) are the leading cause of morbidity, disability-adjusted life years, and mortality among NCDs, causing approximately 17.9 million deaths annually. Generally, individuals affected by CVDs, including coronary heart disease, cerebrovascular disease, and rheumatic heart disease, present elevated blood pressure, obesity, hyperglycemia, and hyperlipidemia, all of which, in combination, cause a metabolic syndrome [3]. Among the modifiable risk factors for CVDs, the most relevant include an unhealthy diet (rich in fats, salt, and sugar, and low in fruits and vegetables), physical inactivity, tobacco use, and excessive alcohol consumption [1].

Considering that a third of all ischemic heart diseases arises from high cholesterol levels, statin drugs are one of the chief therapeutic agents for treating hypercholesterolemia and preventing CVDs [4], which are competitive inhibitors of 3-hydroxy-3-methylglutaryl-coenzyme A (HMG-CoA) reductase, an enzyme that catalyses the key step in the mevalonate pathway [5]. The crystallographic structure of the catalytic portion of human HMG-CoA reductase complexed with HMG, CoA, and NADP+ [6], and statins [7] was established in the early 2000s, and it provides atomic details of the activity of statins binding to human HMG-CoA reductase and how the latter enables the steps of isoprenoids and sterols biosynthesis. Since the hydrophobic groups of statins occupy the HMG-binding pocket, blocking the substrate’s access to it, this class of drugs behaves as competitive inhibitors of HMG-CoA reductase with respect to the substrate, but not to NADPH [7]. It is known that HMG-CoAR increased gene expression, following the statin treatment; attenuates the cholesterol-lowering effect of statins; and increases the risk for side effects. A complementary approach to alleviate this problem has been reported [8]. Furthermore, widespread statin use has raised concerns regarding potential statin-related adverse events (SRAEs), among which myalgia, stiffness, cramps, muscle weakness, and rhabdomyolysis, occurring at either normal or slightly elevated creatine kinase (CK) levels, have been reported in 7–29% of patients. These SRAEs account for up to 75% of treatment discontinuations [9]. In addition, statins have been proposed to inhibit mitochondrial respiratory chain complexes, enhance the formation of reactive oxygen species in this organelle, and induce mitochondrial apoptosis [10]. In fact, it has been suggested that mitochondrial mechanisms may be involved in a variety of non-muscle SRAEs [9].

Long-term statin therapy has been implicated in new-onset diabetes mellitus cases (0.5–1.0%) and haemorrhagic strokes (0.05–0.10%) [11]. In addition to the fact that SRAEs are neither rare nor of trivial impact [9], older adults seem to be more susceptible and less resilient to SRAEs and drug–drug interactions than younger adults [12].

Several studies have proposed that bioactive peptides, particularly those of legume seed origin, represent a possible alternative to statin drugs [13,14,15,16,17], as well as other natural compounds (salvianolic acid, curcumin, and docosanol) [18]. The cholesterol-lowering properties of some of these peptides have been demonstrated in vitro and in vivo [19,20,21]. However, the mechanisms by which they exert these effects have not been fully elucidated. Nevertheless, inhibition of endogenous cholesterol synthesis has been suggested as a putative mechanism of action [19,22]. Peptides appear to interact within the catalytic site of HMG-CoA reductase, resulting in induced fit and its consequent loss of activity via a statin-like mechanism [20,21,23].

Significant decreases in the plasma concentrations of total cholesterol and triacylglycerides were observed in our previous studies in rats when the animals were treated for 28 days with daily doses of β-vignin protein isolated from cowpea [13]. Other studies have also supported the hypocholesterolemic effect of cowpeas [24].

In this study, we employ in silico approaches to investigate the binding mechanism between peptides derived from cowpea β-vignin and HMG-CoA reductase, and applied the obtained information to guide the synthesis and evaluation of one peptide that was found to inhibit the enzyme in vitro via a statin-like mechanism.

## 2. Results and Discussion

### 2.1. Peptide-Binding Sites within HMG-CoA Reductase In Silico

Previous studies have revealed that peptides consisting of three to five amino acid residues are likely to interact within the catalytic site of HMG-CoA reductase [23]. Moreover, di- and tripeptides are readily absorbed by enterocytes via PepT1 transport and are less susceptible to hydrolysis by plasma enzymes during transport; both of these factors increase their chances of reaching the hepatocytes intact [25]. However, larger peptides, such as IAVPGEVA and IAVPTGVA obtained from soybean glycinin [26], and NALEPDNRIESEGG, NALEPDNRIES and PFVKSEPIPETNNE from pigeon peas [19], also bind to the catalytic site of 3-hydroxy-3-methylglutaryl-CoA reductase. These peptides also modulate cholesterol metabolism by activating the LDLR-SREBP2 pathway, which increases LDL uptake by HepG2 cells [19,26]. Nevertheless, peptides consisting of four or more amino acids are prone to hydrolysis, creating dipeptides and tripeptides [27].

As shown in Table 1, in silico, hydrolysis of β-vignin produced 82 peptide sequences, of which nearly 50% were di- and tripeptides. Certain physicochemical features favour peptide binding to the active site of HMG-CoA reductase [20,23]. For instance, we employed aggregation-prone regions (APR) to identify peptide sequences that bind to HMG-CoA reductase [28]. Overall, peptide hydrophobicity has also been shown to predict their affinity for the catalytic region of HMG-CoA reductase [20]. Similarly, hydrophobic interactions play a key role in statin binding to HMG-CoA reductase, despite the structural diversity of these compounds [7].

The hydrophobicity of the peptides ranged from −3.9999 to 4.1500, and hydrophobic residues, such as phenylalanine (19.5% and 6%), tyrosine (8.5%, and 0%), or leucine (30.5%, and 0%), were more frequent at the C-terminus region than at the N-terminus, as the parenthetical percentages show. In addition, several peptides contained aromatic rings with characteristics similar to those reported for statins, as shown in Table 1. The presence of aromatic rings in several of these residues may favour their binding to the catalytic region of HMG-CoA reductase, as they possibly participate in the same interactions as the decalin ring of simvastatin [6]. According to the PeptideRanker Score (PRS) server, 13 peptides (QGF, NF, IAF, GQNNPF, GR, QDF, MPNY, PIY, DSDF, GHL, DVF, QSDSHF, and PHHADADF) provided sufficiently high scores to be considered bioactive (PRS ≥ 0.50) (Table 2). Only one (7.7%) of the predicted peptides contained more than six amino acid residues, whereas ten (77%) had MWs smaller than 500 Da. Previous studies have established that smaller peptides are more likely to display inhibitory activity [17,21]. With the exception of Gly-Arg, all peptides contained at least one aromatic amino acid residue in their sequence, which is considered critical for binding to HMG-CoA reductase [20,28].

Despite indirect evidence that these 13 peptides possibly bind to the HMG-CoA reductase active site, there is no structural or biochemical evidence to support this hypothesis (Table 2). Hence, performing docking studies in either the substrate (HMG-CoA) or in the cofactor (NADPH) binding sites would constitute a biased choice. It has been argued that blind docking can be employed to address this conundrum, but Ghersi & Sanchez [29] have shown that focused docking (to a previously selected binding pocket) provides more accurate docking poses than blind-docking and requires less computational time. This last factor is particularly relevant for peptide docking, because the number of rotational bonds significantly impacts the time required to perform the docking. For this reason, it has been recommended to conduct at least 10 million energy evaluations per trial when peptides are blindly docked to their targets and perform at least 100 trials [30]. Even though blind docking can be performed on online servers (e.g., Achilles Blind Docking Server. Available online: https://bio-hpc.ucam.edu/achilles/ accessed on 23 February 2021), state-of-the-art software, such as Quick Vina-W (Scripps Research Institute, San Diego, CA, USA) [31], still provide success rates (RMSD < 2 Å) below 50% for the best ranked pose. For the reasons stated above, we decided to employ an orthogonal approach, which relies on the chemical similarity between the peptides and ligands that have affinity to the substrate binding site or the cofactor binding site, to outline the most probable binding site for cowpea β-vignin virtually derived peptides. Traditionally, molecules can be compared using 2D (e.g., the MACCS key and 2D hashed fingerprints) and 3D descriptors (e.g., shape, 3D pharmacophore, among others). Other methods rely on molecular superposition instead of quantitative comparisons of the descriptor. Morphological similarity is a ligand-based method that superposes the putative ligands (peptides) on a template molecule (i.e., NADH) in its bioactive conformation, using the putative binding profile of the template as a guide [32]. NADH was employed as the template for molecules that should interact within the cofactor binding site, while simvastatin, atorvastatin, and rosuvastatin were chosen as templates for molecules with high affinity to the substrate binding site.

As expected, the majority of peptides provided lower scores for NADH than for HMG-CoA reductase inhibitors that bind to the substrate binding site (Table 2). The only exception was PHHADADF. Case-by-case analysis revealed that the peptide with the highest morphological similarity for NADH (4.58) was GHL, whereas PYI ranked the best for SIM (6.23), QDF for OTO (5.61), and QGF for ROS (5.42). The lack of consensus among the scores is a possible consequence of the considerable structural diversity of HMG-CoA reductase inhibitors and could be considered a clouding variable for this analysis. To overcome this issue, one might consider only the peptides with high similarity scores for at least two HMG-CoA reductase inhibitors. This approach highlighted QGF, IAF, QDF, and PIY as peptides with the highest probability of interacting within the substrate binding site of HMG-CoA reductase. This result supports a focused docking study of these peptides for the substrate binding site of HMG-CoA reductase.

Several docking software packages are available for docking ligands and/or peptides to the protein; among them, AutoDock Vina (version 4.2.6, Scripps Research Institute, San Diego, CA, USA) is a free option that requires little input from the user and provides reasonable solutions (poses) at low computational costs [33]. Although there are general guidelines for setting the available box size [34], we decided to investigate the influence of the search space on the software success rate. To accomplish this goal, simvastatin was redocked to HMG-CoA reductase (PDB ID: 1HW9) using varying box sizes. Other than a slight increase in the time required for the software to run, no significant change was observed for the best ranked pose of simvastatin, which provided an RMSD <2.0 Å in all cases. As shown in Figure 1a, the predicted simvastatin pose was comparable to the crystallographic binding structure (RMSD = 1.47 Å) when a box size of 10 Å × 10 Å × 10 Å was employed. This set of parameters ensures that the decalin ring, carboxyl, and one of the hydroxyl moieties perform the expected molecular interactions (Figure 1b). As the selected peptides and the ligand had similar molecular masses, the optimised box size was employed to investigate the putative peptide/HMG-CoA reductase binding profile.

Docking software scoring functions are known to be poorly correlated with ligand affinity, due to the simplistic treatment of solvation and entropic effects [33]. Thus, we decided to visually analyse the three best-ranked poses and discard those that had a clear unrealistic binding profile (e.g., Tyrosine OH is buried in a hydrophobic pocket).

Accordingly, the two best-ranked poses for QGF show unfavourable interactions within the binding site and, in the third pose, the phenylalanine side chain is pointed toward the solvent (Figure 1d–f). The best-ranked pose for IAF is predicted to mimic the interactions of the decalin ring of simvastatin and H-bond with Ser565. However, its carboxyl moiety points to Glu559 (Figure 1g–i). Although the two best-ranked poses of QDF displayed similar issues, the third best-ranked pose (Figure 1j–l) presents a reasonable binding profile, wherein phenylalanine (F) occupies the same region as the decalin ring and glutamine (Q) engages in polar interactions, similar to the carboxyl moiety of simvastatin (H-bonding to Lys692, rather than to Arg590). Although the aspartic residue (D) of the peptide explores a novel pocket in the HMG-CoA reductase binding site, its hydrogen network requirements are fulfilled (H-bonding to Ser589). PIY displayed an unfavourable binding profile within the HMG-CoA reductase binding site (Figure 1m–o). The best-ranked pose places the proline pyrrolidine ring within the same pocket as the nonpolar moiety of simvastatin; however, both the tyrosine and isoleucine moieties are loosely positioned in the binding site.

### 2.2. ADMET Parameters

Drug pharmacokinetic (PK) properties, namely absorption, distribution, metabolism, and excretion, as well as drug toxicity, are directly related to their physicochemical and structural properties. PK properties are generally comprehensively evaluated during the final stages of drug development; thus, numerous compounds with poor PK properties require re-optimisation [35]. Therefore, predicting PK properties in the early phases of drug discovery is crucial for guiding the optimisation of a hit-to-lead compound. Considering that IAF and QDF are the most promising peptides, their ADMET parameters (Table 3) were predicted in silico and compared to the most commonly used statins worldwide (simvastatin and atorvastatin) [36]. IAF and QDF are more likely to be absorbed in the gastrointestinal tract owing to their higher Caco-2 cell permeability (80.1% and 93.9%, respectively) than simvastatin and atorvastatin (55% and 82.6%, respectively). On the other hand, human intestinal absorption of the peptides was lower (IAF = 75.9% and QDF = 50.8%) than that of the statins (97.7% and 96.7%, respectively). However, only the physicochemical parameters of the peptides are considered in this approach, while di-, tri-, and short peptides can be absorbed into intestinal epithelial cells (duodenum, jejunum, ileum, and colon) by peptide transporter 1 (PepT1) in the apical membrane of enterocytes, which has a high transport capacity and facilitates efficient intestinal uptake of short peptides [37]. Therefore, intestinal absorption of IAF and QDF is likely to be higher than predicted by this tool.

According to our in silico prediction, both IAF and QDF inhibit OATP and P-glycoprotein (P-gP) at moderate-to-high levels. The inhibition of OATP1B1, OATP2B1, and OATP1B3 transporters results in decreased absorption and distribution of drugs that are substrates of these transporters. This occurs because these drug uptake transporters are localised in the apical membrane of various tissues, mainly in the hepatocytes [38]. Such changes in drug absorption are responsible for a variety of drug–drug interactions (DDIs) and have to be taken into account when using these peptides. Similarly, the modulation of an efflux transporter, such as P-gP, would have a direct impact on the absorption and distribution of drugs that are its substrate, being the majority of drugs on the market. P-gP inhibition leads to enhanced absorption and distribution of a substrate [39]. P-gP is distributed in various organs and tissues, such as the liver, kidney, heart, and lungs, and acts as a defence mechanism against xenobiotics in humans. Our results also indicated that the probability of IAF and QDF being P-gP substrates is high (56% to 70%). Therefore, these peptides can act as both inhibitors and substrates of these efflux transporters and their absorption and tissue distribution is expected to be favourable following oral administration at high concentrations [40]. A similar profile was predicted for statins (Table 3).

The blood–brain barrier (BBB) permeability of IAF (91.6%) and QDF was predicted to be slightly lower than that calculated for simvastatin (98.2%), which does not exhibit central-nervous-system (CNS)-related side effects. The results obtained for mitochondrial subcellular distribution (IAF = 53.8% and QDF = 68%) were also lower than those suggested for simvastatin (73.8%) and atorvastatin (71.5%). Although statins can be absorbed by the gastrointestinal tract, their bioavailability is limited due to extensive first-pass metabolism. Our results suggest that IAF and QDF are not good CYP3A4 substrates, especially when compared to statins, but they have a high probability of being metabolised by CYP2D6 (IAF = 80.4% and QDF = 78.6%). CYP2D6 is the third most important metabolising enzyme of market drugs [41].

The hepatotoxicity effect of statins is well described in the literature [42]. It is important to highlight that hepatotoxicity was predicted to be equal or lower for the peptides (IAF = 57.5% and QDF = 65%) than for atorvastatin (65%). Likewise, prediction of carcinogenic effects indicated that IAF and QDF are safer than statins.

Finally, plasma protein binding (PPB) prediction results (Table 3) suggested that the free plasma concentration of IAF and QDF would be higher than that of statins, increasing their chances of being distributed to the target site and exerting a therapeutic effect [43]. Water solubility (logS) prediction supported this hypothesis, as the peptides were predicted to be less soluble in water than the statins. Although promising, the results described above require experimental validation using cell models.

### 2.3. HMG-CoA Reductase Enzyme Inhibition In Vitro

β-vignin was purified by gel chromatography to a purity of ≥95% (data not shown), following a procedure described in the literature [44]. Then, the protein was hydrolysed by the sequential action of pepsin and pancreatin to obtain peptides with sequences similar to those resulting from the gastrointestinal process [45].

The chromatographic profiles of the total protein hydrolysate (TPH) and its fraction, containing peptides >30 kDa, peptides of 30 to 10 kDa, peptides of 10 to 3 kDa, and peptides smaller than 3 kDa resulting from the in vitro hydrolysis of cowpea β-vignin, are shown in Figure 2. Although a large number of peptides were produced by this protocol, the TPH contained a large number of peptides (more than 30 peaks) (Figure 2a). The small number of peaks in the chromatographic profile of the 3 kDa peptide fraction allowed for its full characterisation by mass spectroscopy (data not shown), whereupon it was confirmed that IAF, QGF, and QDF were among the five major components.

Previous studies have pointed out the difficulty in identifying bioactive peptides derived from enzymatic proteolysis because of the great variability of the primary sequences found, especially for the smaller fragments containing 2–6 amino acid residues [21,46]. Thus, in this study, we used in silico tools to predict the peptide sequences that would interact with HMG-CoA and, consequently, exert a hypocholesterolemic effect. Considering the results from the in silico approach described above and the reported cholesterol-lowering properties of peptides, which likely result from interactions within the catalytic site of HMG-CoA reductase [20], the inhibitory profiles of the hydrolysed protein fractions of cowpea β-vignin were evaluated in vitro (Figure 3a).

The TPH exhibited 95.3% HMG-CoA reductase activity compared to the control (no inhibition) (Figure 3a). The <3 kDa peptide fraction was deemed to be the principal factor responsible for this biological activity, as it inhibited 90% of HMG-CoA reductase activity at the same concentration (*p* < 0.001). In contrast, the >30, 30–10, and 10–3 kDa peptide fractions displayed significantly lower inhibition (12% to 59%) than the <3 kDa peptide fraction or pravastatin (*p* < 0.001), all the same in the 5000 µg/mL concentration. These results differed slightly from those reported by Marques et al. [24], who claimed that cowpea bean peptides with a molecular mass below 3 kDa reduced HMG-CoA reductase activity by 89%. This difference in values could be explained by the fact that a hydrolysate obtained from the total protein isolate of cowpea beans was used in their study, instead of the purified β-vignin protein. However, the fraction with peptides <3 kDa has been shown to be responsible for the effects observed in experimental trials with a variety of approaches [17,21,23,46,47].

In order to verify whether these three peptides are responsible for the HMG-CoA reductase inhibitory activity, the IAF, QGF, and QDF were synthesised and their hypocholesterolemic activity was evaluated (Figure 3b). The chromatographic profiles of the synthesised crude IAF and QGF peptides, following their isolation and characterisation (*m*/*z* ratio), are shown in Appendix A. QDF parameters have been reported in a previously published study [21]. The mass/charge ratios of the synthetic peptides were comparable to the theoretical values, as shown in Appendix A.

The results of the HMG-CoA reductase inhibition assays (Figure 2b) indicated that, at 500 µM concentration, synthetic IAF, QGF, and QDF reduced the enzymatic activity by 69% (*p* < 0.001), 77% (*p* < 0.001), and 78% (*p* < 0.001), respectively, when compared to the control group (no inhibition). Hence, these results are in good agreement with the docking studies previously described, according to which IAF, QGF and QDF act as competitive inhibitors that bind within the active site of HMG-CoA reductase (Figure 1). This predicted binding profile offers a reasonable mechanism for the peptides’ cholesterol-lowering effect and blockage of isoprenoids and sterols biosynthesis [5]. However, one cannot exclude an indirect mechanism of action, such as the modulation of AMP-activated protein kinase phosphorylation of HMGR, which reduces the catalytic efficiency of the enzyme by phosphorylation of serine872 [6], since other peptides were shown to follow this mechanism of action [19,26]. The design and synthesis of QDF and QGF analogues are currently being performed in order to improve their potency; however, this goal has not yet been achieved (data not shown).

Leu-Pro-Tyr-Pro-Arg and Leu-Pro-Tyr-Pro were among the first peptides considered to exert hypocholesterolemic effects [17]. Although several authors claim that direct HMG-CoA reductase inhibition is the main cholesterol-lowering mechanism of action of peptides isolated from soybeans and cowpea beans [16,21,26,46,48], one must consider that other pathways might also contribute to their overall effect. These additional mechanisms of action are currently being investigated. Other peptides derived from glycinin, a soybean protein, such as Ile-Ala-Val-Pro-Gly-Glu-Val-Ala, have also been reported to decrease HMG-CoA reductase activity in vitro. Nevertheless, Lammi et al. [16] proposed a detailed mechanism of action for IAVPGEVA and LPYP. In experiments performed using HepG2 cells, these peptides interfered with the catalytic activity of HMG-CoA reductase and modulated cholesterol metabolism through the activation of the LDLR–SREBP2 pathway, increasing the cell uptake of LDL. In addition, it was observed that this mechanism was involved in the activation of AMPK and ERK 1/2. For the first time, Silva et al. [21] related the Gln-Asp-Phe peptide derived from cowpea β-vignin to the inhibition of HMG-CoA reductase activity (IC_50_ = 12.8 μM) in a dose-dependent manner. The replacement of Asp by Gly does not significantly affect the biological activity (compared with QGF and QDF), but departure from this pattern is detrimental to biological activity. Recently, Kumar et al. [19] have shown that the NALEPDNRIESEGG, NALEPDNRIES, and PFVKSEPIPETNNE peptides inhibit the HMG-CoA reductase, through docking interactions, which were confirmed by in vitro experiments, where the HMGCR inhibitory effect of nearly 70%, 50% and 40%, respectively, in 250 μM concentration was shown. In this study, the authors demonstrated LDL uptake enhancement after treatments with peptides in the extracellular environment of HepG2 cells, suggesting that the mRNA and protein expression of transcription factors are important mechanisms to be explored to determine the cholesterol-lowering therapeutic potential of bioactive peptides. 

Therefore, the results showed in the present study allow the possibility of carrying out further studies with the peptides derived from cowpea β-vignin to act in cholesterol synthesis decrease.

## 3. Materials and Methods

### 3.1. In-Silico Studies

#### 3.1.1. Hydrolysis of the Cowpea β-Vignin Primary Sequence

UniProtKB: the A8YQH5–VIGUN code of the primary sequence of cowpea β-vignin was searched in the NCBI (National Center for Biotechnology Information) database (https://blast.ncbi.nlm.nih.gov/Blast.cgi, accessed on 15 June 2020). Simulated digestion of the cowpea β-vignin protein was performed on the BIOPEP server: http://www.uwm.edu.pl/biochemia/index.php/pl/biopep, accessed on 4 September 2021 (BIOPEP-UWM database, Olsztyn, Województwo warmińsko-mazurskie, Poland), using default parameters and by the sequential action of pepsin (EC 3.4.23.1), trypsin (EC 3.4.21.4), and chymotrypsin (EC 3.4.21.1).

#### 3.1.2. Molecular Features and Bioactivity Prediction

The length, composition, and localisation parameters of each virtually derived β-vignin peptide were determined using the BIOPEP server. The molecular weight (MW), isoelectric point, and hydrophobicity (kcal/mol) were calculated using Protein Property Analysis Software (version 1.1, ProPAS^®^, Beijing Proteome Research Center, Beijing, China), available on http://bioinfo.hupo.org.cn/tools/ProPAS/propas.htm, accessed on 4 September 2021. The net charge and ring parameters were calculated using the PepDraw tool (PepDraw^®^, New Orleans, LA, USA), available on the Tulane University server http://www.tulane.edu/~biochem/WW/PepDraw/index.html, accessed on 4 September 2021, and the lipophilicity (LogP) was calculated using Mcule.com, an online drug discovery platform (Mcule.com^®^, Budapest, Hungary), as available on the server https://mcule.com/apps/property-calculator/, accessed on 4 September 2021. The bioactivity probabilities of β-vignin-derived peptides were analysed according to the PeptideRanker^®^ Score (PRS) (Shields’ Lab, Dublin, Ireland), as available on the server http://bioware.ucd.ie/~compass/biowareweb/, accessed on 4 September 2021. In the sequence, peptides with a score ≥0.500 (range of 0.000–1.000) were selected for subsequent analysis related to the interaction of peptides with protein surfaces.

#### 3.1.3. Prediction of Putative Peptide Binding Sites in HMG-CoA Reductase

To investigate whether the virtually derived β-vignin peptides bind to the active site or cofactor binding site of HMG-CoA reductase, the chemical similarity of each peptide to known HMG-CoA reductase inhibitors (simvastatin extracted from 1HW9, rosuvastatin extracted from 1HWL, and atorvastatin extracted from 1HWK) and to the enzyme cofactor (NADH) were calculated using SURFLEX-SIM software (BioPharmics LLC, Santa Rosa, California, USA). The 2D structures of all the peptides were drawn using Marvin Sketch and then energy minimized with a Tripos force field (convergence criteria 0.01 Kcal/mol and dielectric constant = 80), as available on the SYBYL-X 2.0 platform. Then, each peptide was flexibly aligned to the crystallographic coordinates of mevastatin, rosuvastatin, atorvastatin, and NADH using SURFLEX-SIM default parameters. To reduce the “root mean square deviation” (RMSD) among molecules, pre-run and post-run minimisation options were employed. The software provides a similarity score ranging from 0.0 (0.0% similarity) to 10.0 (100% similarity). The top four best-ranked peptides according to the similarity score for at least two ligands had their interaction profile with HMG-CoA reductase predicted by molecular docking.

#### 3.1.4. Molecular Docking

The interaction profiles of the selected peptides were predicted by molecular docking using the AutoDock Vina^®^ software (version 4.2.6, Scripps Research Institute, San Diego, California, USA) [49]. First, water molecules and ligands from the 1HW9 PDB file were discarded and hydrogen atoms were added in random orientations, as available on the SYBYL 2.1-X software. Then, the PDB file was converted to PDBQT format using AutoDock tools [50]. Next, the peptides and simvastatin were sketched in 2D format and then energetically minimised using the single-point optimised AM1 semi-empirical method (Keywords: 1SCF XYZ ESP NOINTER SCALE = 1.4 NSURF = 2 SCINCR = 0.4 NOMM), as implemented in the MOPAC module available on the SYBYL 2.1-X software. Then, the search space was centred on simvastatin crystallographic coordinates (X = 3.93, Y = −9.20, Z = −11.33, PDB ID: 1HW9), and the influence of box size (10 Å to 14 Å cubic box) on redocking accuracy was investigated. The box size that provided the lowest RMSD for redocked simvastatin was selected for molecular peptide docking. The four best-ranked poses were visually analysed in PyMOL (version 0.99r) and those that presented unrealistic interactions with HMG-CoA reductase were discarded.

#### 3.1.5. Absorption, Distribution, Metabolism, Excretion, and Toxicity (ADMET) Parameters

The chemical structures of the selected peptides had their ADMET parameters predicted and compared to those of statins (simvastatin and atorvastatin) using the ADMET SAR^®^ software, version 2.0 (ADMET, Yonsei Engineering Research Complex, Yonsei University, Seoul, Korea), as available on the server https://preadmet.bmdrc.kr/adme-prediction/, accessed on 4 September 2021.

### 3.2. In Vitro Experiments

#### 3.2.1. Preparation of Peptide Fractions with Varying Molecular Sizes from Cowpea β-Vignin

Cowpea (*Vigna unguiculata,* L. Walp) seeds were obtained from the local market (Salvador, BA, Brazil). All reagents were purchased from Sigma–Aldrich^®^ (St. Louis, MO, USA) unless otherwise specified. Isolation procedures and subsequent extensive purification by chromatography (molecular exclusion and ion-exchange) and SDS-PAGE electrophoresis (Hoefer MiniVE electrophoresis system, Amersham Biosciences^®^, Hercules, CA, USA) of β-vignin were performed as previously described. The cowpea flour was homogenized with 1/20 (*w*/*v*) 0.1 mol/L in distilled water, which was adjusted to pH 7.5 with 2 mol/L NaOH. Stirring was conducted for 1 h at room temperature. The suspension was then centrifuged (10,000× *g* for 40 min at the same temperature). From this step onwards, all following procedures were carried out at 4 °C. The supernatant fraction was diluted 1/1 with distilled water; its pH was adjusted (5.0 with 2 mol/L HCl) and then kept overnight. Another centrifugation allowed us to obtain a soluble fraction (albumins) and a pellet. The pellet was homogenised in water 1:20 (*w* (initial weight)/*v*) using a Potter–Elvehjem homogeniser and stirred (4 °C for 10 min). Then, the pH was adjusted to 5.0 and the suspension centrifuged as above. The new pellet was dissolved in 0.1 mol/L NaCl 1:10 (*w* (initial weight)/*v*) at pH 7.0 and stirred (4 °C for 20 min). After further centrifugation, the pellet was discarded and the supernatant fraction was diluted 1/1 (*v*/*v*) with distilled water; the pH was adjusted to 5.0 and the solution was kept at 4 °C overnight. The latter centrifugation step allowed us to recover a pellet, mainly consisting of β-vignin. The pellet was dissolved in 0.2 mol/L NaCl at pH 7.0, and dialysed before freeze-drying [13].

Aliquots of freeze-dried β-vignin were solubilised in a 50 mM potassium phosphate buffer, pH 7.5, containing 0.5 M NaCl and 0.01% NaN_3_, and fractionated by size exclusion on a Sepharose CL-6B column (1.0 cm × 100 cm), previously equilibrated with the same buffer. The flow rate of the column was kept at 5.8 mL/min. The protein eluted from the column was monitored by measuring absorbance at 280 nm. The main fraction was collected, dialyzed against distilled water, and then lyophilized. The chromatographed β-vignin was dissolved in a 50 mM Tris-HCl buffer, pH 7.5, and fractionated by ion exchange chromatography on a Mono-Q column (5.0 cm × 0.5 cm). A NaCl concentration gradient from 0.05 to 0.5 M lasting forty minutes, with flow rate of 1 mL/min, was applied. To wash the column, a 1.0 M NaCl solution containing 0.01% NaN_3_ was used [44].

A simulated human gastrointestinal digestion was performed according to the sequential hydrolysis protocol [51], with minor modifications [21]. β-vignin was hydrolysed by pepsin (enzyme/substrate ratio 1:66, 37 °C for 3 h, pH = 2), the pH was neutralised, and the hydrolysed β-vignin was further treated with pancreatin (enzyme/substrate ratio 1:25, 37 °C for 3 h, pH = 7) (Sigma–Aldrich^®^, St. Louis, MO, USA). The total protein hydrolysed (TPH) was ultrafiltered with a molecular weight cut-off (MWCO) of 3, 10, and 30 kDa via sequential ultrafiltration using Microcon centrifugal filters ultrafiltration membrane filters (Millipore^®^, Merck, Darmstadt, Germany) to yield peptide fractions of <3, 3–10, 10–30, and >30 kDa, which were tested in the in vitro HMG-CoA reductase inhibitory activity assay. The relative protein concentration was determined using bovine serum albumin as the reference [52].

#### 3.2.2. Profiling of Peptide Fractions by RP-HPLC

The chromatographic profiles of the TPH and its fraction, containing peptides >30 kDa, peptides of 30 to 10 kDa, peptides of 10 to 3 kDa, and peptides smaller than 3 kDa, were determined by high performance liquid chromatography (HPLC), using a PerkinElmer System with a reverse phase column (C18 × 0.45 × 25 cm) and an UV/VIS detector (PerkinElmer^®^, Waltham, MA, USA). The gradient used was 10 min in 95% A and 50 min to achieve 25% B. The solvent system used was A (0.045% trifluoroacetic acid in ultrapure water) and B (0.036% trifluoroacetic acid in acetonitrile), with a flow rate of 1.0 mL/min, temperature of 30 °C, and readings recorded at 220 nm [21].

#### 3.2.3. Peptide Synthesis

The selected peptides were synthesised by a solid-phase synthesis using Fmoc (9-fluorenylmethyloxycarbonyl) amino acid standards as proposed by the authors of [53]. The crude peptides were purified by a semi-preparative HPLC (Shimadzu Prominence^®^, Kyoto, Japan) using a reversed phase (25 cm × 10 mm × 10 μm) column (Jupiter Proteo Phenomenex^®^, Torrance, CA, USA) combined with a UV/VIS detector. The solvent system used was A (0.045% trifluoroacetic acid in ultrapure water) and B (0.036% trifluoroacetic acid in acetonitrile). The gradient used was as follows: 15 min in 5% B and 90 min to achieve 45% B at a flow rate of 5 mL/min. The degree of purity was analysed by RP-HPLC and the peptides were identified on the basis of the ESI-MS data (LC/ESI-MS, Bruker^®^ Amazon, Billerica, MA, USA). The solvent system used was A (0.1% formic acid in ultrapure water) and B (0.1% formic acid in acetonitrile), with a flow rate of 0.5 mL/min at a temperature of 30 °C. The gradient used was 5–95% of B in 20 min, as previously established [21].

#### 3.2.4. HMG-CoA Reductase Enzyme Assay

The HMG-CoA reductase inhibitory activities of the cowpea peptide fractions and synthesised peptides were assayed using the HMG-CoA reductase kit (Sigma–Aldrich^®^, St. Louis, MO, USA). The TPH sample and its fractions, containing peptides > 30 kDa, peptides of 30 to 10 kDa, peptides of 10 to 3 kDa, and peptides smaller than 3 kDa, were tested and diluted in aqueous buffer at the final concentration of 5000 µg/mL. QGF, IAF, and QDF peptides were diluted in the same buffer at the final concentration of 500 µM, following the manufacturer’s instructions. The peptide concentration was calculated based on the measured molecular mass of each one of them (351.2 g/mol, 350.2 g/mol, and 408.2 g/mol, respectively). No aggregation was observed in the hydrolysate, peptide fractions, or synthesized peptides. Enzyme activity was measured by monitoring the decrease in absorbance at 340 nm and 37 °C, following the supplier’s instructions. Specific enzyme activity was defined as number of micromoles of oxidised NADPH/min/mg protein (mgP). The results are expressed as the percentage of the control-specific activity of the enzyme in the absence of pravastatin and the compounds under investigation. Activity was calculated using the following equation: Units/mgP = (∆A_340_/min_sample_ − ∆A_340_/min_blank_) × TV/12.44 × V × 0.6 × LP, where 12.44 = ε^mM^; the extinction coefficient for NADPH at 340 nm is 6.22 mM^−1^ cm^−1^, hence 12.44 represents 2 NADPH units consumed in the reaction; TV = total volume of the reaction (mL); V = volume of the enzyme used in the assay (mL); 0.6 = enzyme concentration in mgP/mL; and LP = light path in cm.

### 3.3. Statistical Analysis

In vitro results were expressed as the mean ± standard deviation (SD) of at least three independent analyses. The means of the results were evaluated using a one-way analysis of variance (ANOVA) at a significance level of 5%. Tukey’s multiple-range test was used for multiple comparisons (SigmaStat, v. 3.5, Systat software, San Jose, CA, USA). Statistical significance was set at *p* ≤ 0.05.

## 4. Conclusions

Employing in silico and in vitro experiments, we established that Ile-Ala-Phe, Gln-Gly-Phe, and Gln-Asp-Phe peptides are capable of inhibiting HMG-CoAR activity via a competitive statin-like mechanism. Molecular docking studies indicated that the peptides have a higher affinity to the substrate binding site than to the NADH binding site of HMG-CoAR. The peptides inhibited HMG-CoAR activity in a manner akin to the interactions of the decalin ring of simvastatin and via H-bonding. In all cases, phenylalanine was necessary for this interaction. The pharmacokinetic profile of the peptides suggested that the chances of achieving the desired concentration in the human body would be higher for the peptides than for the statins. In vitro inhibition studies indicated that the presence of peptides in the TPE derived from cowpea β-vignin resulted in interference with HMG-CoAR activity, especially the < 3 kDa peptide fraction. It was confirmed that Ile-Ala-Phe, Gln-Gly-Phe, and Gln-Asp-Phe peptides inhibited the catalytic activity of HMG-CoAR by 69%, 77%, and 78%, respectively. Although our results suggest Ile-Ala-Phe, Gln-Gly-Phe, and Gln-Asp-Phe peptides derived from cowpea β-vignin have the potential to lower cholesterol synthesis via a statin-like regulation mechanism, their oral bioavailability has not been measured in this work. Furthermore, animal experiments or cell-based assays are required to fully assess the impact of those peptides on cholesterol control.

## Figures and Tables

**Figure 1 ijms-22-11067-f001:**
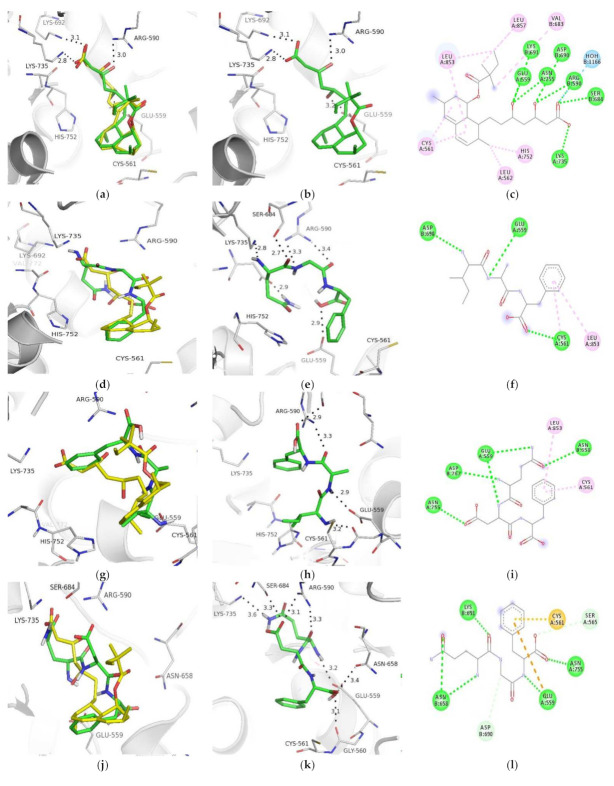
Docking analysis of simvastatin and the peptides with the HMG-CoA reductase catalytic complex (PDB ID: 1HW9) using the AutoDock Vina software. (**a**) Best-ranked pose for simvastatin (yellow); (**b**) the crystallographic binding profile of simvastatin (grey) in the HMG-CoA reductase active site; and (**c**) 2D diagram of simvastatin interactions. Orientations of QGF, IAF, QDF, and PIY docked into the HMG-CoA reductase active site are shown in comparison to the bioactive simvastatin conformation in (**d**,**g**,**j**,**m**), respectively; their predicted binding profile in the active site is shown in (**e**,**h**,**k**,**n**), respectively; and the 2D diagram of the peptides interactions is shown in (**f**,**i**,**l**,**o**), respectively. The protein structure is depicted with selected residues highlighted as sticks (carbon = light grey) and ligands are represented as sticks (carbon = yellow or grey; oxygen = red; and nitrogen = blue). All images were generated using PyMOL 0.99r software (Scripps Research Institute, San Diego, CA, USA).

**Figure 2 ijms-22-11067-f002:**
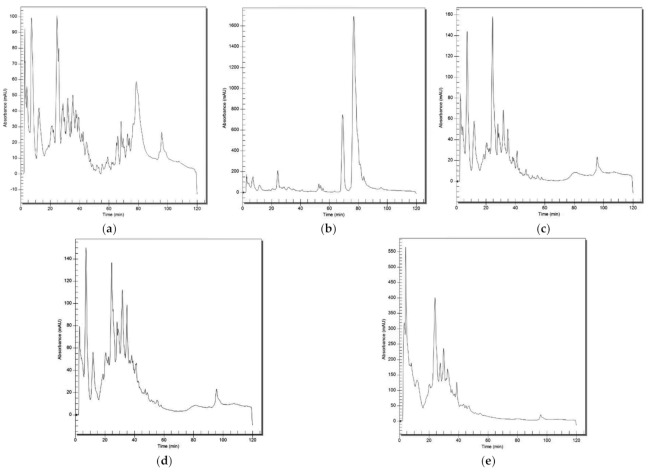
RP-HPLC chromatogram of the (**a**) total hydrolysed extract, (**b**) fraction containing peptides > 30 kDa, (**c**) peptides of 30 to 10 kDa, (**d**) peptides of 10 to 3 kDa, and (**e**) peptides smaller than 3 kDa. See text for experimental details.

**Figure 3 ijms-22-11067-f003:**
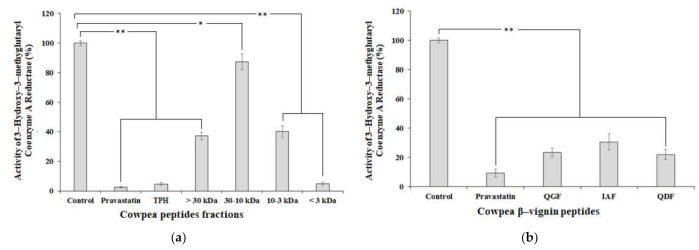
Inhibition of 3-hydroxy-3-methylglutaryl-coenzyme A reductase (HMG-CoAR) activity by (**a**) peptide fractions and (**b**) peptides on 5000 µg/mL and 500 µM concentrations, respectively. Mean value was significantly different: * *p* ≤ 0.05 and ** *p* ≤ 0.001 by Tukey’s multiple-range test. See text for experimental details.

**Table 1 ijms-22-11067-t001:** Molecular characteristics of peptides derived from cowpea β-vignin via sequential hydrolysis in silico.

N°	Sequence ^a^	N° aa ^a^	Localisation ^a^	MW ^b^	Isoelectric Point ^b^	Hydrophobicity (Kcal/mol) ^b^	Net Charge ^c^	Ring ^c^	PRS ^d^
1	VPL	3	1–3	327.42	5.49	2.1333	0	1	0.37
2	GVL	3	7–9	287.36	5.52	2.5333	0	0	0.31
3	ASL	3	12–14	289.33	5.57	1.6000	0	0	0.25
4	SVSF	4	15–18	438.48	5.24	1.3500	0	1	0.41
5	GIVHR	5	19–23	580.69	9.76	0.1200	+1	1	0.20
6	GHQESQEESEPR	12	24–35	1412.39	4.48	−2.6916	−3	2	0.09
7	GQNNPF	6	36–41	675.70	5.52	−1.6166	0	2	0.80
8	DSDR	4	44–47	491.46	4.21	−3.0749	−1	0	0.20
9	HTL	3	50–52	369.42	6.74	−0.0332	0	1	0.14
10	NQY	3	55–57	423.43	5.52	−2.7666	0	1	0.16
11	GHL	3	58–60	325.37	6.74	0.0667	0	1	0.59
12	VL	2	62–63; 202–203	230.31	5.49	4.0000	0	0	0.13
13	QR	2	64–65	302.33	9.75	−3.9999	+1	0	0.27
14	DQR	3	67–69	417.42	5.84	−3.8332	0	0	0.19
15	SK	2	70–71; 184–185	233.27	8.47	−2.3499	+1	0	0.07
16	QIQNL	5	72–76	614.70	5.52	−0.4399	0	0	0.23
17	ENY	3	77–79	424.41	4.00	−2.7666	−1	1	0.07
18	VVEF	4	81–84	492.57	4.00	1.9250	−1	1	0.13
19	QSK	3	85–87	361.40	8.75	−2.7332	+1	0	0.07
20	PNTL	4	88–91	443.50	5.96	−0.4999	0	1	0.32
21	PHHADADF	8	94–101	908.93	5.05	−1.0749	−2	4	0.51
22	VVL	3	104–106	329.44	5.49	4.0667	0	0	0.06
23	NGR	3	107–109	345.36	9.75	−2.7999	+1	0	0.46
24	AIL	3	110–112	315.41	5.57	3.3667	0	0	0.33
25	TL	2	113–114	232.28	5.19	1.5500	0	0	0.14
26	VNPDGR	6	115–120	656.70	5.81	−1.5499	0	1	0.39
27	DSY	3	121–123	383.36	3.80	−1.8666	−1	1	0.18
28	IL	2	124–125	244.33	5.52	4.1500	0	0	0.39
29	EQGHAQK	6	126–132	796.84	6.85	−2.3142	0	1	0.08
30	TPAGTTF	7	133–139	693.75	5.19	0.0714	0	2	0.33
31	VNHDDNENL	9	142–150	1069.05	4.02	−1.7999	−3	1	0.15
32	IVK	3	152–154	358.48	8.75	1.6000	+1	0	0.05
33	AVPVNNPHR	9	156–164	1003.13	9.80	−0.8555	+1	3	0.47
34	QDF	3	166–168	408.41	3.80	−1.3999	−1	1	0.74
35	SSTEAQQSY	9	171–179	999.99	4.00	−1.4555	−1	1	0.08
36	QGF	3	181–183	350.37	5.52	−0.3666	0	1	0.94
37	NIL	3	186–188	358.44	5.52	1.6000	0	0	0.26
38	EASF	4	189–192	452.46	4.00	0.0750	−1	1	0.30
39	DSDF	4	193–196	482.45	3.56	−1.2499	−2	1	0.60
40	EINR	4	198–201	530.58	6.10	−1.7499	0	0	0.08
41	GEEEQK	6	205–210	718.72	4.25	−3.0499	−2	0	0.04
42	QQDEESQQEGVIVQL	12	211–225	1729.82	3.50	−1.0666	−4	0	0.11
43	EQIR	4	228–231	544.61	6.10	−1.7499	0	0	0.09
44	EL	2	232–233	260.29	4.00	0.1500	−1	0	0.07
45	MK	2	234–235	277.38	8.50	−0.9999	+1	0	0.45
46	HAK	3	236–238	354.41	8.76	−1.7666	+1	1	0.11
47	STSK	4	239–242	421.45	8.47	−1.5499	+1	0	0.06
48	SL	2	244–245	218.25	5.24	1.5000	0	0	0.33
49	STQNEPF	7	246–252	821.84	4.00	−1.5428	−1	2	0.35
50	NL	2	253–254	245.28	5.52	0.1500	0	0	0.29
51	SQK	3	256–258	361.40	8.47	−2.7332	+1	0	0.08
52	PIY	3	259–261	391.47	5.95	0.5333	0	2	0.60
53	SNK	3	262–264	347.37	8.47	−2.7332	+1	0	0.09
54	GR	2	266–267	231.25	9.75	−2.4499	+1	0	0.77
55	HEITPEK	7	269–275	852.94	5.40	−1.6999	−1	2	0.09
56	NPQL	4	276–279	470.53	5.52	−1.1999	0	1	0.47
57	DL	2	281–282	246.26	3.80	0.1500	−1	0	0.33
58	DVF	3	283–285	379.41	3.80	1.1667	−1	1	0.55
59	TSVDIK	6	287–292	661.75	5.50	−0.0332	0	0	0.08
60	EGGL	4	293–296	374.39	4.00	−0.1249	−1	0	0.37
61	MPNY	4	298–301	523.60	5.27	−1.1249	0	2	0.74
62	NSK	3	302–304	347.37	8.75	−2.7332	+1	0	0.07
63	AIVIL	5	305–309	527.70	5.57	3.7600	0	0	0.24
64	VVNK	4	310–313	458.56	8.72	0.2500	+1	0	0.03
65	GEANIEL	7	314–320	744.80	3.79	−0.1142	−2	0	0.14
66	VGQR	4	321–324	458.52	9.72	−1.0499	+1	0	0.17
67	EQQQQQQEESW	11	325–335	1447.44	3.67	−3.0181	−3	2	0.09
68	EVQR	4	336–339	530.58	6.10	−1.8249	0	0	0.04
69	AEVSDDDVF	9	342–350	996.00	3.37	−0.1999	−4	1	0.19
70	VIPASY	6	351–356	648.76	5.49	1.1333	0	2	0.17
71	PVAITATSNL	10	357–366	986.13	5.96	0.8800	0	1	0.14
72	NF	2	367–368;381–382	279.30	5.52	−0.3499	0	1	0.94
73	IAF	3	369–371	349.43	5.52	3.0333	0	1	0.82
74	GINAENNQR	9	372–380	1015.05	6.00	−1.7888	0	0	0.12
75	AGEEDNVMSEIPTEVL	16	384–399	1732.88	3.45	−0.2562	−5	1	0.16
76	DVTF	4	400–403	480.52	3.80	0.7000	−1	1	0.32
77	PASGEK	6	404–409	587.63	6.43	−1.3999	0	1	0.17
78	VEK	3	410–412	374.44	5.97	−1.0666	0	0	0.02
79	INK	3	414–416	373.45	8.75	−0.9666	+1	0	0.10
80	QSDSHF	6	417–422	719.71	5.08	−1.4999	−1	2	0.52
81	TDHSSK	6	423–428	673.68	6.41	−2.1499	0	1	0.06
82	EER	3	430–432	432.43	4.53	−3.8332	−1	0	0.04

To predict cleavage sites of cowpea-bean 7S-globulin, the amino acid sequence available in the UniProt database was used (Identity: A8YQH5_VIGUN). As described in the Materials and Methods section. ^a^ The data for peptides, isoelectric point, molecular weight (MW), amino acid number, and repetition were accessed from BIOPEP (http://www.uwm.edu.pl/biochemia/index.php/pl/biopep, accessed on 4 September 2021). ^b^ Hydrophobicity value for each peptide was calculated using the following equation: Peptide hydrophobicity = ∑ [(molecular weight of each amino acid) × (substituent constant, characterising this amino acid side chain)]/∑ (molecular weight of each amino acid), using ProPAS^®^ version 1.1 software, available at http://www.mybiosoftware.com/propas-1-1-protein-property-analysis-software.html, accessed on 4 September 2021. ^c^ Peptide net charge and ring were calculated and predicted using the PepDraw tool, available at http://www.tulane.edu/~biochem/WW/PepDraw/index.html, accessed on 4 September 2021 ^d^ The PeptideRanker score (PRS) refers to the probability of peptide bioactivity, available at http://bioware.ucd.ie/~compass/biowareweb/Server_pages/peptideranker.php, accessed on 4 September 2021.

**Table 2 ijms-22-11067-t002:** Peptides derived from the cowpea β-vignin protein and their predicted interactions with the HMG-CoA reductase catalytic site.

N°	Sequence ^a^	PRS ^b^	Lipophilicity (LogP)	SURFLEX-SIM ^c^			
				NADH	SIM	ATO	ROS
1	QGF	0.94	−0.478	4.28	6.18 *	5.39 *	5.42 *
2	NF	0.94	0.094	3.33	5.42	4.44	4.24
3	IAF	0.82	−0.555	3.83	6.01	5.57 *	4.91 *
4	GQNNPF	0.80	0.533	3.52	4.28	4.70	4.06
5	GR	0.77	0.689	3.53	5.37	3.95	4.41
6	QDF	0.74	−1.089	4.30	6.09 *	5.61 *	4.36
7	MPNY	0.74	1.897	3.97	4.86	4.32	4.29
8	PIY	0.60	−1.859	4.05	6.23 *	4.95	5.08 *
9	DSDF	0.60	1.223	3.96	5.27	5.27	5.07 *
10	GHL	0.59	1.056	4.58	5.75	4.81	4.50
11	DVF	0.55	0.793	3.58	6.09 *	5.52 *	4.67
12	QSDSHF	0.52	2.159	3.45	4.15	4.70	3.63
13	PHHADADF	0.51	−1.990	3.64	3.00	2.94	2.58

^a^ Peptide sequences were obtained from BIOPEP analysis, as described in the Materials and Methods Section. ^b^ PRS = PeptideRanker score (see Table 1). ^c^ Similarity score calculated by SURFLEX-SIM, ranging from 0.0 to 10.0, that compares the interaction profile of each peptide with that expected for HMG-CoA reductase inhibitors or for the enzyme cofactor. Compounds with higher similarity scores for HMG-CoA reductase inhibitors than for the enzyme cofactor suggest that they have a higher probability of binding within the catalytic binding site than to the cofactor binding site. Peptides that scored among the top four (*), for at least two ligands were subjected to molecular docking studies.

**Table 3 ijms-22-11067-t003:** Comparison of absorption, distribution, metabolism, excretion, and toxicity properties of the peptides and statins.

ADMET Parameters	IAF	QDF	SIM	ATOR
**Absorption**	Probability	Probability	Probability	Probability
Human oral bioavailability	0.5429	0.5857	0.9571	0.9196
Caco-2	0.8014	0.9394	0.5503	0.8264
Human Intestinal Absorption	0.7591	0.5081	0.9767	0.9669
OATP2B1 inhibitor	1.0000	1.0000	1.0000	0.6023
OATP1B1 inhibitor	0.9011	0.9287	0.7740	0.8473
OATP1B3 inhibitor	0.9466	0.9505	0.9480	0.9237
**Distribution**	Probability	Probability	Probability	Probability
P-glycoprotein inhibitor	0.7977	0.6699	0.9198	0.7590
P-glycoprotein substrate	0.5570	0.7014	0.9344	0.6434
Blood–Brain Barrier (BBB)	0.9164	0.9524	0.9822	0.5857
Mitochondrial subcellular distribution	0.5377	0.6796	0.7384	0.7153
**Metabolism**	Probability	Probability	Probability	Probability
CYP3A4 substrate	0.5726	0.5225	0.7410	0.6733
CYP2C9 substrate	0.5943	0.5648	1.0000	0.7914
CYP2D6 substrate	0.8036	0.7860	0.8893	0.7542
CYP inhibitory activity	0.9732	0.9845	0.8682	0.7663
**Toxicity**	Probability	Probability	Probability	Probability
Carcinogenicity (binary)	0.6571	0.7143	0.9286	0.8143
Carcinogenicity (trinary)	0.6963	0.7645	0.7060	0.4690
Hepatotoxicity	0.5750	0.6500	0.9000	0.6500
**ADMET predicted profile Regressions**	**Unit**	**Unit**	**Unit**	**Unit**
Water solubility (logS)	−2.319	−1.964	−5.483	−3.813
Plasma protein binding (%)	0.770	0.595	0.918	0.878

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
