# Peer review of "IAF, QGF, and QDF Peptides Exhibit Cholesterol-Lowering Activity through a Statin-like HMG-CoA Reductase Regulation Mechanism: In Silico and In Vitro Approach"

_ijms, 2021, doi:10.3390/ijms222011067_

Round 1

Reviewer 1 Report

ijms-1389885_comments

The authors described the tripeptides IAF, QGF, and QDF as promising inhibitors of HMG-CoA reductase (HMG-CoAR) based on the docking experiments and the HMG-CoAR activity assay and believe that these peptides have the potential to lower the cholesterol levels.

  • In Fig. 2; what is the control used in experiments; mention in the figure caption; the labels for samples (a, b, c…) and the figure labels (a) and (b) are confusing; maybe use different labels.
  • line 261; should mention “synthetic IAF, QGF, and QDF reduced the enzymatic activity by 69% …”
  • The elution profiles for synthetic and digested peptides are entirely different. In Fig. 3, the chromatogram has many peaks/peptides. Authors should mention which HPLC fractions are considered for testing
  • What are the solvents used in peptide purification
  • Sample preparation details for both crude peptide fractions and synthetic peptides - should be provided
  • Should mention how the peptide concentration was measured
  • The quality of figures should be improved: in Fig. 1, some of the images (I, O) are pulled in one direction; the text font is too small to read
  • The mass-spec data on the samples from Fig. 3 can be included in the Supplementary Information
  • Why are HPLC/mass-spec profiles for QDR not provided?
  • Should mention the final peptide yields
  • Should mention the solubility of the synthetic peptides (any aggregation observed for these peptides in aqueous solution/buffer?)
  • Conclusions should appear after the discussion

Author Response

Response to reviewers

Dear Editor

Please find enclosed the revised version of the manuscript entitled “IAF, QGF, and QDF peptides exhibit cholesterol-lowering activity through a statin-like HMG-CoA reductase regulation mechanism: In silico and in vitro approach”. We would like to thank the reviewers for their criticisms and considerations, which contributed positively to the scientific quality of the manuscript. All comments and issues raised by the referee were dealt with and resulted in the modifications described below and highlighted in the manuscript

Reviewer 1 comments:

Point 1 – In Fig. 2; what is the control used in experiments; mention in the figure caption; the labels for samples (a, b, c…) and the figure labels (a) and (b) are confusing; maybe use different labels.

Answer: As requested by the reviewer, the figure 3 (before figure 2) caption was modified to improve its clarity (Page 9, lines 232-235).

Point 2 – Line 261; should mention “synthetic IAF, QGF, and QDF reduced the enzymatic activity by 69% …”

Answer: In the sentence was added term “synthetic” (Page 12, line 262 to 263).

Point 3 – The elution profiles for synthetic and digested peptides are entirely different. In Fig. 3, the chromatogram has many peaks/peptides. Authors should mention which HPLC fractions are considered for testing

Answer: The point raised by the reviewer was taken into consideration and resulted in the following modification in the manuscript to improve its clarity that the eluition profiles of the fraction containing peptides (Figure 2) and synthetic peptides (Figure S4) are otherwise.

Page 10, line 231 to 233 – “The chromatographic profiles of the total protein hydrolysate (TPH) and its fraction containing peptides > 30 kDa, peptides 30 to 10 kDa, peptides 10 to 3 kDa, and peptides smaller than 3 kDa resulting from the in vitro hydrolysis of cowpea β-vignin, are shown in Figure 2. Although a large number of peptides were produced by this protocol. The TPH contained a large number of peptides (more than 30 peaks) (Fig. 2a).”

Page 15, line 403 to 409

“4.2.3 Peptide fractions profiling by RP-HPLC

The chromatographic profiles of the TPH, fraction containing peptides > 30 kDa, peptides 30 to 10 kDa, peptides 10 to 3 kDa, and peptides smaller than 3 kDa were determined by high performance liquid chromatography (HPLC), using a PerkinElmer System with a reverse phase column (C18×0.45×25 cm) and an UV/VIS detector. The gradient used was 10 min in 95% A and 50 min to achieve 25% B. The solvent system used was A (0.045% trifluoroacetic acid in ultrapure water) and B (0.036% trifluoroacetic acid in acetonitrile), with a flow rate of 1.0 ml/min, temperature of 30 °C, and readings recorded at 220 nm[22]”

Page 15, line 411 to 420

4.2.3 Peptide synthesis

The selected peptides were synthesised by solid-phase synthesis using Fmoc (9-fluorenylmethyloxycarbonyl) amino acid standards as proposed by [55]. The crude peptides were purified by semi-preparative HPLC (Shimadzu Prominence®, Kyoto, Japan) using a reversed phase (25 cm × 10 mm × 10 μm) column (Jupiter Proteo Phenomenex®, CA, USA) combined with a UV/VIS detector. The solvente system used was A (0.045% trifluoroacetic acid in ultrapure water) and B (0.036% trifluoroacetic acid in acetonitrile). The gradient used was as follows: 15 min in 5% B, and 90 min to achieve 45% B at a flow rate of 5 mL/min., and the degree of purity was analysed by RP-HPLC and the peptides were identified on the basis of the ESI-MS data (LC/ESI-MS, Bruker® Amazon). The solvent system used was A (0.1% formic acid in ultrapure water) and B (0.1% formic acid in acetonitrile), with a flow rate of 0.5 mL/min. at a temperature of 30 °C. The gradient used was as follows: 5–95% of B in 20 min., as previously established[22]

Point 4 – What are the solvents used in peptide purification

Answer: The information about the solvents used in peptide purification was added in the manuscript to improve its clarity (Page 15, line 411 to 420)

“The crude peptides were purified by semi-preparative HPLC (Shimadzu Prominence®, Kyoto, Japan) using a reversed phase (25 cm × 10 mm × 10 μm) column (Jupiter Proteo Phenomenex®, CA, USA) combined with a UV/VIS detector. The solvente system used was A (0.045% trifluoroacetic acid in ultrapure water) and B (0.036% trifluoroacetic acid in acetonitrile). The gradient used was as follows: 15 min in 5% B, and 90 min to achieve 45% B at a flow rate of 5 mL/min., and the degree of purity was analysed by RP-HPLC and the peptides were identified on the basis of the ESI-MS data (LC/ESI-MS, Bruker® Amazon). The solvent system used was A (0.1% formic acid in ultrapure water) and B (0.1% formic acid in acetonitrile), with a flow rate of 0.5 mL/min. at a temperature of 30 °C. The gradient used was as follows: 5–95% of B in 20 min., as previously established[22]”

Point 5 – Sample preparation details for both crude peptide fractions and synthetic peptides - should be provided

Answer: The details about the crude peptide fractions and synthetic peptides were added, as requested by the reviewer.

Page 14, line 374 to 393

“Briefly, the cowpea flour was homogenized with 1/20 (w/v) 0.1 mol/L in distilled water, adjusted to pH 7.5 with 2 mol/L NaOH. Stirring was applied for 1 h at room temperature. The suspension was then centrifuged (10 000 g for 40 min at the same temperature). From this step, all following procedures were carried out at 4°C. The supernatant fraction was diluted 1/1 with distilled water; its pH was adjusted (5.0 with 2 mol/L HCl) and then kept overnight. A further centrifugation step as above allowed us to obtain a soluble fraction (albumins) and a pellet. The pellet was homogenised in water 1:20 (w (initial weight)/v) by using a Potter-Elvehjem homogeniser and stirred (4°C for 10 min). Then, the pH was adjusted to 5.0 and the suspension centrifuged as above. The new pellet was dissolved in 0.1 mol/L NaCl 1:10 (w (initial weight)/v) at pH 7.0 and stirred (4°C for 20 min). After further centrifugation as above, the pellet was discarded and the supernatant fraction was diluted 1/1 (v/v) with distilled water; the pH was adjusted to 5.0 and the solution was kept at 4°C overnight. The latter centrifugation step allowed us to recover a pellet, mainly consisting of β-vignin. The pellet was dissolved in 0.2 mol/L NaCl at pH 7.0, and dialysed before freeze-drying.

Aliquots of freeze dried β-vignin were solubilised in 50 mM potassium phosphate buffer, pH 7.5, containing 0.5 M NaCl and 0.01% NaN3, and fractionated by size exclusion on Sepharose CL-6B column (1.0 cm × 100 cm), previously equilibrated with the same buffer. The flow rate of the column was kept at 5.8 mL/min. The protein eluted from the column was monitored by measuring absorbance at 280 nm. The main fraction was collected, dialyzed against distilled water, and then lyophilized. The chromatographed β-vignin was dissolved in 50 mM Tris-HCl buffer, pH 7.5 and fractionated by ion exchange chromatography on Mono-Q column (5.0 cm × 0.5 cm). A NaCl concentration gradient from 0.05 to 0.5 M lasting forty minutes, with flow rate of 1 mL/min, was applied. To wash the column, 1.0 M NaCl solution containing 0.01% NaN3 was used”

Page 15, line 412 to 420

“The selected peptides were synthesised by solid-phase synthesis using Fmoc (9-fluorenylmethyloxycarbonyl) amino acid standards as proposed by [55]. The crude peptides were purified by semi-preparative HPLC (Shimadzu Prominence®, Kyoto, Japan) using a reversed phase (25 cm × 10 mm × 10 μm) column (Jupiter Proteo Phenomenex®, CA, USA) combined with a UV/VIS detector. The solvente system used was A (0.045% trifluoroacetic acid in ultrapure water) and B (0.036% trifluoroacetic acid in acetonitrile). The gradient used was as follows: 15 min in 5% B, and 90 min to achieve 45% B at a flow rate of 5 mL/min., and the degree of purity was analysed by RP-HPLC and the peptides were identified on the basis of the ESI-MS data (LC/ESI-MS, Bruker® Amazon). The solvent system used was A (0.1% formic acid in ultrapure water) and B (0.1% formic acid in acetonitrile), with a flow rate of 0.5 mL/min. at a temperature of 30 °C. The gradient used was as follows: 5–95% of B in 20 min., as previously established [48].”

Point 6 – Should mention how the peptide concentration was measured

Answer: The information about the peptide’s concentration was added in the manuscript to improve its clarity.

Page 16, line 424 to 430

“The HMG-CoA reductase inhibitory activities of the cowpea peptide fractions and synthesised peptides were assayed using the HMG-CoA reductase kit (Sigma Aldrich®, St. Louis, MO, USA). TPH sample, fractions containing peptides > 30 kDa, peptides from 30 to 10 kDa, peptides from 10 to 3 kDa and peptides smaller than 3 kDa were tested diluted in aqueous buffer at the final concentration of 5,000 µg/mL, and QGF peptides, IAF and QDF were diluted in the same buffer at the final concentration of 500 µM, following the manufacturer's instructions. The peptides concentration was calculated based on the measured molecular mass of each one of them, being considered 351.2 g/mol, 350.2 g/mol, and 408.2 g/mol, respectively.”

Point 7 – The quality of figures should be improved: in Fig. 1, some of the images (I, O) are pulled in one direction; the text font is too small to read

Answer: The quality all the figures was adjusted as requested by the reviewer.

Point 8 – The mass-spec data on the samples from Fig. 3 can be included in the Supplementary Information

Answer: Adjusted as requested by the reviewer (Page 16, line 445 to 446).

Point 9 – Why are HPLC/mass-spec profiles for QDF not provided?

Answer: These informations were published in a previous study by our research group, as mentioned in manuscript.

Page 11, line 258 to 261

“The chromatographic profiles of the synthesised crude IAF and QGF peptides following their isolation and characterisation (m/z ratio), are shown in Fig. S4. QDF parameters have been reported in previously published study[22]. The mass/charge ratios of the synthetic peptides were comparable to the theoretical values, as shown in Table S4.”

Point 10 – Should mention the final peptide yields.

Answer: Information on experimental data regarding peptide synthesis has been added to the manuscript (Table S4).

Page 11, line 260 to 261“The mass/charge ratios of the synthetic peptides were comparable to the theoretical values, as shown in Table S4.”

Page 17, line 250 to 251“Table S4. Experimental data on peptide synthesis”

Point 11 – Should mention the solubility of the synthetic peptides (any aggregation observed for these peptides in aqueous solution/buffer?)

Answer: The information “no aggregation was observed” has been added.

Page 16, line 429 to 430 “No aggregation was observed in the hydrolysate, peptide fractions or synthesized peptides.”

Answer: The information “no aggregation was observed” has been added.

Point 12 – Conclusions should appear after the discussion

Answer: Adjusted as requested by the reviewer (Page 13, line 300 to 312)

We hope the revised version of the manuscript meets Journal of Functional Foods publication criteria.

Sincerely.

Professor Ederlan de Souza Ferreira

College of Pharmacy

Federal University of Bahia

147 Barão de Jeremoabo street, Ondina, Salvador, BA, Brazil.

[email protected] 

Reviewer 2 Report

Atherosclerosis can trigger cardiovascular disease which is the main cause of death worldwide. This condition is an essential reason for conducting studies related to alternative medicines from natural resources to reduce plasma cholesterol levels. Nowadays, a widely-used treatment to reduce high cholesterol levels utilizes statins. It is a drug but it has some adverse side effects. Therefore, other alternative compounds that are more effective and safer than statin to patients who are resistant to or intolerant of conventional pharmacotherapy is required.

The authors described the research methods well, described the results correctly and held a discussion.

In my opinion, the introduction should include information about other natural compounds for the treatment of hypercholesterolaemia. For example, Pandanus tectorius fruits extract rich in tangeretin, ethyl caffeate, and coffeic guinic acid et al.

Authors should also renumber the tables and add Table 1S to the text of the manuscript (not supplementary).

Author Response

Response to reviewers

Dear Editor

Please find enclosed the revised version of the manuscript entitled “IAF, QGF, and QDF peptides exhibit cholesterol-lowering activity through a statin-like HMG-CoA reductase regulation mechanism: In silico and in vitro approach”. We would like to thank the reviewers for their criticisms and considerations, which contributed positively to the scientific quality of the manuscript. All comments and issues raised by the referee were dealt with and resulted in the modifications described below and highlighted in the manuscript

Reviewer 2 comments:

Point 1 – In my opinion, the introduction should include information about other natural compounds for the treatment of hypercholesterolaemia. For example, Pandanus tectorius fruits extract rich in tangeretin, ethyl caffeate, and coffeic guinic acid et al.

Answer: We fully agree with the reviewer opinion, and we recognize this fact must be made crystal clear for International Journal of Molecular Sciences’ readers.

Page 13, line 300 to 312 – “Several studies have proposed that bioactive peptides, particularly those of legume seed origin, represent a possible alternative to statin drugs[14–18], as well as other natural compounds (Salvianolic acid, curcumin, and docosanol)[19]”

Point 2 – Authors should also renumber the tables and add Table 1S to the text of the manuscript (not supplementary).

Answer: Adjusted as requested by the reviewer (Page 3, line 94 to 95)

We hope the revised version of the manuscript meets Journal of Functional Foods publication criteria.

Sincerely.

Professor Ederlan de Souza Ferreira

College of Pharmacy

Federal University of Bahia

147 Barão de Jeremoabo street, Ondina, Salvador, BA, Brazil.

[email protected] 
